# Dietary Long-Chain Fatty Acids and Cognitive Performance in Older Australian Adults

**DOI:** 10.3390/nu11040711

**Published:** 2019-03-27

**Authors:** Lesley MacDonald-Wicks, Mark McEvoy, Eliza Magennis, Peter W. Schofield, Amanda J. Patterson, Karly Zacharia

**Affiliations:** School of Health Sciences, Faculty of Health, University of Newcastle, Callaghan, NSW 2308, Australia; mark.mcevoy@newcastle.edu.au (M.M.); eliza.magennis@uon.edu.au (E.M.); peter.schofield@newcastle.edu.au (P.W.S.); amanda.patterson@newcastle.edu.au (A.J.P.); karly.zacharia@uon.edu.au (K.Z.)

**Keywords:** fatty acid, cognitive performance, Hunter Community Study (HCS), Mini Mental State Examination (MMSE), Audio Recorded Cognitive Screen (ARCS)

## Abstract

Convincing evidence exists for the positive effect of an improvement in diet quality on age-related cognitive decline, in part due to dietary fatty acid intake. A cross-sectional analysis of data from the Hunter Community Study (HCS) (*n* = 2750) was conducted comparing dietary data from a validated Food Frequency Questionnaire (FFQ) with validated cognitive performance measures, Audio Recorded Cognitive Screen (ARCS) and Mini Mental State Examination (MMSE). Adjusted linear regression analysis found statistically significant associations between dietary intake of total n-6 fatty acids (FA), but no other FAs, and better cognitive performance as measured by the ARCS (RC = 0.0043; *p* = 0.0004; *R*^2^ = 0.0084). Multivariate regression analyses of n-6 FA intakes in quartiles showed that, compared with the lowest quartile (179.8–1150.3 mg), those in the highest quartile (2315.0–7449.4 mg) had a total ARCS score 2.1 units greater (RC = 10.60466; *p* = 0.006; *R*^2^ = 0.0081). Furthermore, when n-6 FA intake was tested against each of the ARCS domains, statistically significant associations were observed for the Fluency (RC = 0.0011432; *p* = 0.007; *R*^2^ = 0.0057), Visual (RC = 0.0009889; *p* = 0.034; *R*^2^ = 0.0050), Language (RC = 0.0010651; *p* = 0.047; *R*^2^ = 0.0068) and Attention (RC = 0.0011605; *p* = 0.017; *R*^2^ = 0.0099) domains, yet there was no association with Memory (RC = −0.000064; *p* = 0.889; *R*^2^ = 0.0083). No statistically significant associations were observed between FA intakes and MMSE. A higher intake of total n-6 FA, but not other types of FA, was associated with better cognitive performance among a representative sample of older aged Australian adults.

## 1. Introduction

The rapidly increasing prevalence of an age-related decline in cognitive performance is problematic across the globe, with an estimated 44 million people currently living with dementia worldwide [1]. Approximately 430,000 Australian’s were estimated to have dementia in 2018 [1]. This is expected to rise, reflecting the age structure and growth of the Australian population, with an estimation of 4.0% of the population to have dementia by the year 2050 [2]. Not only does age-related cognitive decline have a large impact on individuals, affecting quality of life and increasing neuropsychiatric symptoms and disability, it also places a significant financial burden upon society [3,4,5]. In 2016, the estimated annual total cost of dementia was $8.8 billion within Australia alone, an increase of 81% is predicted by 2036 to an estimated $25.8 billion [6]. Delaying the onset of cognitive impairment would decrease the average number of years spent living with the disease, and consequently reduce healthcare costs [4]. Therefore, examining cognitive performance, the factors that may affect it and finding ways to prevent or decrease age-related cognitive impairment is a public health imperative for Australia.

Epidemiological studies support the hypothesis that dietary quality and pattern can influence an individual’s cognitive performance throughout their life-cycle [7]. Fatty acid and fish intake in particular have been shown to have an effect on cognitive performance [8,9,10,11]. The Mediterranean Diet (MeDi) includes a wide range of foods and food groups, and therefore a range of fatty acids, including n-9 monounsaturated fatty acid (MUFA), n-3 and to a lesser extent n-6 polyunsaturated fatty acids (PUFA), and saturated fatty acids (SFA). It is the most studied dietary pattern to date and has been convincingly linked with lower rates of cognitive decline [12]. A systematic review with meta-analysis of prospective cohort studies found a 13% decreased risk of dementia with adherence to the MeDi (RR = 0.87; 95% CI: 0.81, 0.94; *p* ≤ 0.00001) [13]. Another systematic review of randomised controlled trials (RCTs) and observational studies supported these findings and concluded that a higher adherence to the MeDi was associated with lower rates of cognitive decline, better cognitive function, and reduced risk of Alzheimer’s disease (AD) [14].

In light of the above findings, it is important to identify those elements within the MeDi—which includes fish, fruit, vegetables, nuts and seeds—that contribute alone, or in combination, to the cognitive benefits that derive from the MeDi as a whole.

Fish may constitute a key ingredient of the MeDi. A meta-analysis of cohort studies by Wu et al. (2015) found that a higher intake of fish was associated with a 36% lower risk of AD (95% CI 8–51%) [15], and the addition of 100 g of fish per week was associated with an 11% lower AD risk (RR = 0.89, 95% CI: 0.79–0.99) [15]. Several longitudinal population-based studies provide further evidence that frequent fish consumption may be related to slower rates of cognitive impairment and better global cognitive function [8,9,10]. While these studies provide good evidence for the positive effects of fish consumption, a recent study by Van De Rest et al. (2009) concluded that a higher dietary intake of fish measured by FFQ, had no association with cognitive performance in a sample of 1025 elderly men [11]. Although not reporting a statistical significance, the fish intake in this study was notably lower than the previous supportive studies (median intake range 0.2–4.2 servings per week) [11].

With the overall effects of fish consumption on cognitive performance reported to be positive [8,9,10], it is notable that long chain omega 3 polyunsaturated fatty acids (LCn3PUFA), found in high concentration in many fish, has been associated with decreased risk of AD and mild cognitive impairment (MCI), and overall better cognitive aging [12,16,17,18,19,20,21], although, in one recent meta-analysis of cohort studies, the association did not achieve statistical significance [15].

Interestingly, the positive effects of dietary LCn3PUFA intake may not translate into supplementation trials, as a Cochrane review of randomised controlled trials (RCTs) found no benefit of LCn3PUFA on cognitive function in older individuals without dementia [22]. Evidently, existing literature on the role of LCn3PUFA on cognitive performance is inconsistent, and it is worth considering that the positive effects seen for the MeDi might be related to other fatty acids (FAs) both in the MeDi and in fish. Mono-unsaturated Fatty Acids (MUFAs) are thought to be another protective component of the MeDi, where they are consumed primarily in the form of olive oil [19]. A population-based, prospective study by Solfrizzi et al. (2006) showed that total MUFA intake (median intake of 41.87 g/day at baseline) was not associated with incident MCI, when measured with the mini mental state examination (MMSE) [23]. However, a study of equal power and similar design undertaken in older aged women found that a higher total MUFA consumption (median intake of 19.39 g/day) was associated with better cognitive function when tested by a neuropsychological test battery [24]. In regard to n-6 fatty acids, a prospective study by Van Gelder et al. (2007) found a positive association between high linoleic acid (18:2 n-6) PUFA intake and cognitive impairment in a cohort of 476 men aged 69–89 years [9]. Another study by Kalmijn et al. (1997) found that higher linoleic acid was associated with lower risk of dementia, although this was not statistically significant (RR = 0.6, 95% CI: 0.3–1.2) [25]. Research on the roles of both MUFA and n-6 fatty acids is limited and lacks consistency and therefore cannot be used to devise an evidence base, highlighting the need for further research in the area.

With convincing evidence for positive effects of the MeDi on age related cognitive decline, mostly positive results for fish consumption, and inconsistent results for both dietary and supplemented n-3 intakes, it seems that trying to identify just one type of FA (i.e., LCn3FA) that is responsible for these positive cognitive effects, without considering other fatty acids, is unproductive. More research examining the role of other fatty acids in the MeDi, including n-6 PUFA, MUFA and SFA, on cognitive performance is warranted. Therefore, the aim of the present study was to evaluate whether a range of dietary long-chain fatty acids are associated with cognitive performance in older men and women participating in the Hunter Community Study (HCS) in Newcastle, Australia.

## 2. Methods

### 2.1. Study Sample

The present study was a cross sectional analysis of data from the Hunter Community Study (HCS). The HCS is a cohort study of community-dwelling men and women, aged 55–85 years at baseline who reside in Newcastle, New South Wales (NSW), Australia. It is a collaborative study between the Hunter New England Area Health Service and the University of Newcastle’s School of Medicine and Public Health. The study commenced in 2004 and was established to investigate factors important in the health, well-being, social functioning and economic consequences of ageing. A total of 9784 participants were randomly selected from the NSW electoral roll and received two letters of introduction and an invitation to participate. In total, 3253 people participated in the study, which provided a population profile reflecting that of the Hunter, state and national Australian profiles in terms of both gender and marital status, although with a slightly lower mean age [26]. Participants were required to participate in a range of clinical measurements, including blood samples, anthropometric data, respiratory function, cognition, bone mineral density, functional capacity and cardiovascular performance. They were also required to complete a series of self-administered questionnaires, regarding demographic characteristics, morbidity, medication use, nutrition, mental health status, physical activity, quality of life, daytime sleepiness, social support and occupational exposures [26]. Ethical approval for this research was granted by the University of Newcastle Human Research Ethics committee (H-820-0504).

### 2.2. Food Frequency Questionnaire (FFQ)

Dietary intake at baseline was measured using a 145-item semi-quantitative FFQ [26]. The self-administered questionnaire requires respondents to indicate the usual frequency of consumption of food items, using a nine-category frequency scale ranging from never to four or more times per day, during the previous 12 months. The questionnaire has been validated against four-day weighed food records, in an elderly population [27] Fatty acid intakes were determined from the FFQ, using NUTTAB95 and the Royal Melbourne Institute of Technology Fatty Acid databases [28,29]. Fatty acid variables used in the present analysis included: (i) total n-3 intake (g/day); (ii) total n-6 intake (g/day); (iii) total MUFA intake (g/day); (iv) total PUFA intake (g/day); (v) total SFA intake (g/day); and (vi) total fat intake (g/day).

### 2.3. Cognitive Performance Outcome Measures

The primary outcome variables examined in this study were cognitive performance as measured by the total score on the mini mental state examination (MMSE) and the Audio Recorded Cognitive Screen (ARCS).

The Mini-Mental State Examination (MMSE) is an interview-administered test of cognitive function, concentrating primarily on the cognitive aspects of mental functions. The first component of the test is scored out of 21 and covers orientation, memory, and attention, requiring vocal responses only. The second part is scored out of nine, and tests ability to name, follow verbal and written commands, write a sentence spontaneously, and copy a complex polygon. Responses are converted to a score out of 30, where higher scores indicate better cognitive function and a score of ≤23 signals cognitive impairment [30]. Concurrent validity was determined in the correlation of MMSE scores with the Wechsler Adult Intelligence Scale, verbal IQ scores (Pearson *R* = 0.776; *p* < 0.0001) and performance IQ scores (Pearson *R* = 0.660; *p* <0.001) [30]. The MMSE has satisfactory reliability, with positive results on retesting at both 24 h and 28 days (Pearson *R* = 0.887) [30].

The ARCS is an objective self-administered clinical measure of cognitive performance. It is designed to identify distinct patterns of cognitive impairment through analysis of the major cognitive domains. These key domains are: verbal episodic memory, verbal fluency, visuospatial functioning, language, attention and executive function and speed of writing [31]. A total ARCS score is generated by the sum of scaled scores for all domains and then recalibrated to a population mean of 100 (SD = 15), with a higher ARCS score indicating better cognitive function. The audio testing component of the ARCS is 34 min in duration, is unsupervised and proceeds with a preamble describing the process in a distinct, non-threatening manner. The construct validity of ARCS has been tested against conventional neuropsychological tests probing comparable cognitive domains. Correlations between raw ARCS scores and neuropsychological measures lie mostly within the range *R* = 0.50–0.70, and were all statistically significant (*p* ≤ 0.001) [31]. The ARCS has been validated within a sample of 22 clinic attendees with a repeat (alternate form) assessment over a mean interval of 90 days (range of 24–168 days). Mean reliability coefficients for scaled ARCS domain and global scores were as follows: verbal episodic memory, 0.80; verbal fluency, 0.88; visuospatial functioning, 0.82; language, 0.82; attention and executive function, 0.70; QuickARCS, 0.86; and overall ARCS, 0.88 [31].

### 2.4. Measurement of Potentially Confounding Variables

Potential confounding variables were identified as participant’s age, gender, body mass index (BMI), smoking status, total energy intake, alcohol consumption, physical activity level, education level, diabetes, asthma, hypertension, fish oil supplement use, household income, and carbohydrate, protein, total sugars, cholesterol, and fibre intake. Possible confounders were selected based on existing knowledge of their potential link with cognitive function and diet.

Continuous variables included in the analysis are as follows: macro- and micro-nutrients, as grams or milligrams per day as appropriate; smoking, as packs per year; and alcohol consumption, as mean alcohol intake per drinking day [28]. Education was categorised as primary school only, secondary school not completed, secondary school completed, trade or technical college qualification, and university or other tertiary qualification. Personal income was categorised as either <$40,000 or ≥40,000 per year. Information on self-reported physician-diagnosed co-morbidities were categorised as yes or no to represent presence or absence and included asthma, diabetes and hypertension [28]. Self-reported supplement use (including fatty acids such as fish oil) was also collected.

### 2.5. Statistical Analysis 

All analyses performed were completed using the STATA IC statistical software version 13. To improve the validity of the dietary analyses, participants were excluded from the analysis if reported energy intakes were <4.5 or >20.0 MJ/day, as these were considered biologically improbable and indicative of misreporting [32]. Total n-3 FA, n-6 FA, MUFA, PUFA, and SFA intake measured at baseline were tested for association with baseline cognitive performance—total score of MMSE and ARCS. The means of normally distributed continuous variables were compared between men and women for socio-demographic characteristics and each category of fatty acid. A *p*-value of <0.2 was considered statistically significant when determining potential confounders by testing their statistical association with both cognitive performance and fatty acid intake, based on the definition of a confounder.

Linear regression models were used to examine the association of each fatty acid type mentioned above (grams per day) with cognitive performance outcomes (total MMSE and total ARCS score). A *p*-value of 0.05 was considered statistically significant and regression coefficients, *R*-squared and *p*-values reported as appropriate. Univariate linear regression was used to examine the association between each category of fatty acid and total MMSE and total ARCS score without adjustment for potentially confounding variables. Multivariate linear regression was used to examine the association with adjustment for potential confounding variables.

Variables were also tested in quartile ranges for their association with individual ARCS domain scores to identify specific cognitive domains associated with each fatty acid.

## 3. Results

A total of 3253 older aged men and women participated in the Hunter Community study, 503 (15.5%) of whom were excluded from the current analyses after removal of those with implausible energy intakes. Overall, a total of 2750 participants (1198 men and 1552 women) between the ages of 55 and 85 years were included in the analyses. The demographics of the study population are shown in Table 1.

Mean energy intake for this population was 8334.8 kJ/day (SD = 2353.7), with a mean energy contribution of 18.3% from protein, 45.6% from carbohydrate and 27.0% from fat. Dietary intake data were plausible with regard to energy intakes, and the mean MMSE scores were not dissimilar to population intake data (2011–2012 Australian Health Survey results) (Table 1), therefore making the study meaningful to the population of interest [32,33].

Demographic data were similar between genders, with the exception of smoking status and alcohol intake, which were significantly higher for men (Table 1).

Table 1 includes the mean cognitive outcome scores for both the MMSE and ARCS. The mean score for MMSE was 28.0 (SD = 1.6) and for ARCS, 99.0 (SD = 9.9), with women scoring marginally higher on both scales.

Mean FA variables for both men and women in the HCS cohort are presented in Table 2. The 2011–2012 Australian Health Survey results on FA intakes (for Australian men and women aged 51 years and older) are also presented for comparison [32]. All HCS mean FA results were similar to Australian Health Survey data across comparative age groups. FA intakes from the HCS cohort were comparable between genders, with men consuming slightly greater amounts.

Table 3 and Table 4 show the results of the un-adjusted and adjusted linear regression models for each FA with total ARCS and MMSE scores, respectively. No statistically significant associations were found between dietary FAs and total MMSE score (Table 3).

There was a statistically significant association between total n-6 FA intake and total ARCS score (Adjusted RC = 0.0043; *p* = 0.0004), however there were no statistically significant associations with any other FA type (Table 4).

Categorising FA intakes into quartiles for further multivariate regression analysis did not result in any statistically significant associations with total MMSE score. Similarly, in adjusted analyses there were no statistically significant results observed for quartiles of total fat, PUFA, MUFA, SFA or total n-3 FA with total ARCS score. However, statistically significant results were observed in adjusted analyses between n-6 FAs and total ARCS score. Compared to the lowest quartile of n-6 FA intake (179.8–1150.3 mg), those in the highest quartile (2315.0–7449.4 mg) had a total ARCS score 2.1 units greater (RC = 10.60466; *p* = 0.006; *R*^2^ = 0.0081). When n-6 FA intake was tested against each of the five ARCS domains (Memory, Fluency, Visual, Language and Attention) there were no statistically significant associations for the Memory domain (RC = −0.000064; *p* = 0.889; *R*^2^ = 0.0083), however, statistically significant associations were observed for the Fluency (RC = 0.0011432; *p* = 0.007; *R*^2^ = 0.0057), Visual (RC = 0.0009889; *p* = 0.034; *R*^2^ = 0.0050), Language (RC = 0.0010651; *p* = 0.047; *R*^2^ = 0.0068) and Attention (RC = 0.0011605; *p* = 0.017; *R*^2^ = 0.0099) domains.

## 4. Discussion

The current study aimed to assess the associations between dietary long-chain fatty acids and cognitive performance, in a large cohort of older Australian adults, aged 55-86 years (43.6% men and 56.4% women). The HCS participants provide a population profile that reflects the Hunter (NSW, Australia), state and national Australian profiles in terms of both gender and marital status, however has a slightly lower mean age [26]. The mean MMSE score for men was 27.8 (SD = 1.7) and 28.2 (SD = 1.5) for women. These scores represent good cognitive function, as scores ≤23 are considered to signal cognitive impairment [30]. Mean ARCS scores were reported as 98.3 (SD = 10.1) for men and 99.5 (SD = 9.7) for women, well above what is considered to signify cognitive impairment [31].

Our findings demonstrated that individuals with higher consumption of n-6 FAs had better cognitive performance as measured by the ARCS. This is in contrast to what might be expected as many studies suggest that linoleic acid has an atherogenic role by increasing the oxidative modification of low-density lipoprotein cholesterol, a situation that would theoretically be detrimental to cognitive function [35,36]. Although our results detected a statistically significant relationship between total n-6 intake and ARCS score, the regression coefficient was very small and may not be clinically meaningful (RC = −0.0000113; *p* = 0.0004; *R*^2^ = 0.0084). When broken into quartile ranges of n-6 intake, we found that, compared with the lowest quartile (179.8–1150.3 mg), those in the highest quartile of intake (2315.0–7449.4 mg) had a total ARCS score 2.1 units greater. Further research is needed to determine if those in the highest quartile of intake are less likely to progress to MCI or dementia. To the authors’ knowledge, this is the first examination of dietary intake of total n-6 FA intakes and cognitive performance in a large older aged cohort.

In this cohort, the dietary linoleic acid intake (18:2 n-6) comprised approximately 90% of total n-6 intake. Contrary to much of the existing literature, which shows a positive association between high linoleic acid intake and increased cognitive impairment, our results showed less cognitive impairment with increased intake [9,25,37]. The current recommended linoleic acid intake (AI-Adequate Intake) levels for adults (≥19 years) are 1300 mg/day for men and 800 mg/day for women, which are comparable to the levels observed in the lower quartiles of intake in the present study [29]. It is noted that the set AI does not reflect optimal intake of linoleic acid, but is based on the observed intake in populations with no known fatty acid deficiency [29]. Currently, there is no upper limit (UL) set as there is no known level at which adverse effects may occur [29]. Our findings suggest that as linoleic acid intake is increased the impact on cognitive performance is positive.

The ARCS tool was recently developed and is designed to identify distinct patterns of cognitive impairment through analysis of the major cognitive domains. The total ARCS score can be broken down into five domains of cognitive function, including Memory, Fluency, Visual, Language and Attention. Our results showed no association between total n-6 intake and the Memory domain, however there were statistically significant associations for the other four domains.

Linoleic acid and other n-6 PUFAs are known to be key components of biomembranes and therefore play an important role in cell development, maintenance, function and integrity [38]. During the process of ageing, a reduction of arachidonic acid (20:4 n-6) and adrenic acid (22:4 n-6) concentrations has been observed in the cortex and cerebellum of the brain [39]. The cortex is known to play an important role in many brain functions, including a role in memory, language and attention [40]. Although the cerebellum is most understood for its contribution to motor control, recent studies suggest an important role in cognition, particularly memory, attention, visuospatial functions and language [41]. To the authors’ knowledge, this is the first examination of dietary intake of n-6 FA and specific cognitive domains. Our study supports the need for further research into the effect of n-6 FA intake on cognitive ageing, with particular interest into the effects on separate cognitive domains. It should be acknowledged that the one statistically significant result may be an effect of chance, however further investigations into the effects of n-6 FA intakes on cognitive aging are still warranted.

The positive associations between total n-6 FA intake and cognitive function were not observed when cognitive performance was measured with the MMSE. Furthermore, no statistically significant relationships were observed for any other FA variable and total MMSE score. These findings may be attributed to the differences in the two cognitive tools used in this study. The MMSE is a well-recognised and commonly used cognitive measure in research, particularly survey-based research, however it is best used in a longitudinal setting as it is a good measure of cognitive change over time [42,43]. Due to the cross-sectional design of this study, the MMSE may not be sensitive enough to detect differences in cognitive performance between participants at one time point. It has also been suggested that the MMSE may only be sensitive when a patient is already significantly impaired, and therefore may not be suitable for the high-scoring healthy participants of the HCS [42]. However, the MMSE provided a comparison to the validated ARCS tool, which has been recently developed and shows greater sensitivity [31,44].

Contrary to much of the existing literature, which shows a positive association between increased dietary n-3 FA intake and cognitive performance, the present analyses found no statistically significant associations with the cognitive outcomes used in this study [12,16,17,18,19,20,21]. This inconsistency might be explained by the use of the self-administered FFQ, which has not been validated for the measurement of FA variables. When compared with the Australian Health Survey results, all obtainable FA levels were comparable, however LCn-3PUFA and LCn-6PUFA population intakes were not available for comparison [32]. Given this, our results do support the findings of a recent meta-analysis of cohort studies that reported no statistical evidence for the association between dietary LCn3PUFA and dementia or AD [15]. This suggests that the widely accepted belief for desirable effects of LCn3PUFA on cognitive performance may be unwarranted, and further longitudinal studies within the area are necessary.

Our results show no significant association between total fat, PUFA, MUFA or SFA and cognitive performance when measured by ARCS. There appears to be no consistent evidence base within the existing literature on the effects of total MUFA intake on cognitive function. A prospective cohort study by Naqvi et al. (2011) found that a median intake of 19.39 g/day, similar to intakes observed in HCS, was associated with better cognitive function in a cohort of older aged women [24]. When compared with the current study, tools used to measure cognitive function differed, with participants in the study by Naqvi et al. undertaking an extensive neuropsychological test battery which is likely to show more sensitivity than the ARCS [24]. In regard to SFA intake, once again the results observed in the current study do not reflect the majority of existing literature that states a diet high in SFA is linked with increased cognitive decline and AD disease [10,13,17,21,45]. Of note is that many of these studies reported higher mean intakes of SFA (range = 2430–3230 mg/day) than those observed in the HCS (mean = 1827.6 mg/day), suggesting that the deleterious effects may be limited to individuals with high levels of intake.

The strengths of the present study include the large representative sample, the use of validated dietary assessment and cognitive performance tools, and the ability to adjust for multiple confounding variables. The findings for this study are limited by a potential for measurement error associated with the use of a self-reported FFQ. While this FFQ has been validated for use within an elderly cohort, it has not been validated for the measurement of FA intakes [27]. The use of nutrient databases ensured sensitivity for FA intake but may be limited in assessing the bioavailability of food sources. However, FA levels were realistic when compared with the Australian Health Survey (2013) results after data were limited to plausible energy intake ranges, using a previously published method [28,32]. It must also be noted that dietary data collected from cognitively impaired individuals may be less reliable, due to the increased possibility of either under- or overestimation, leading to differential misclassification [46]. To measure cognitive function, both the MMSE and ARCS tools were used. As previously stated, the MMSE is often used in a longitudinal setting as it best measures cognitive change over time, therefore it may not be the most desirable tool for cognitive measurement in a cross-sectional study [43]. Of course, the cross-sectional nature of this particular study using HCS data, means that causal relationships cannot be inferred, however the findings warrant further investigation using longitudinal analyses to assess temporal associations once further cognitive data become available for HCS participants.

## 5. Conclusions

The current study demonstrated that, among a representative sample of older aged Australian adults, an increased intake of n-6 FA, but no other FA, was associated with better cognitive performance. These findings deviate from the current literature and indicate the need for further studies to clarify the effects of all FAs on cognition using longitudinal study designs.

## Figures and Tables

**Table 1 nutrients-11-00711-t001:** Baseline characteristics for participants from the Hunter Community Study.

	Women		Men		Total	
	(*n* = 1552)		(*n* = 1198)		(*n* = 2750)	
	Mean	SD	Mean	SD	Mean	SD
Age (years)	66.6	7.6	66.8	7.7	66.6	7.7
Weight (kg)	73.1	14.6	86	14	79.3	15.7
Height (cm)	159.5	8.5	172.9	7.1	166	10.2
BMI (kg/m^2^)	28.7	5.6	28.7	4.2	28.7	5
Smoker (packs/year) ^1^	7.6	19.1	15.7	26.6	11.1	23.1
Alcohol intake ^2^	1.2	1.6	3.1	3.6	2	2.8
Total Energy (kJ)	8058.1	2225.7	8693.3	2465	8334.8	2353.7
% Energy from Protein	18.8	3.3	17.6	3	18.3	3.2
% Energy from CHO	47.4	6.4	45.5	6.9	45.6	6.7
% Energy from Fat	27.1	5.1	27	5.2	27	5.1
MMSE Score	28.2	1.5	27.8	1.7	28	1.6
ARCS Score	99.5	9.7	98.3	10.1	99	9.9

^1^ Smoker (packs/year) refers to mean packs of cigarettes smoked per year for all participants. ^2^ Alcohol intake refers to mean intake per drinking day. Abbreviations; BMI, Body Mass Index; CHO, Carbohydrate; MMSE, Mini Mental State Exam; ARCS, Audio Recorded Cognitive screen; SD, Standard Deviation.

**Table 2 nutrients-11-00711-t002:** Mean fatty acid intakes (standard deviation) comparison of the HCS and derived intake from the 2011–2012 Australian Health Survey (AHS) [34].

	HCS Results						AHS Results			
	Women		Men		Total		Women		Men	
	Mean	SD	Mean	SD	Mean	SD	51–70	70+	51–70	70+
Total fat (g)	57.9	20.2	62.5	22.5	59.9	21.3	62	55.6	78.6	66.9
PUFA (g)	8.7	3.5	9.2	3.8	8.9	3.6	9.9	8.8	11.8	9.6
MUFA (g)	19.1	6.9	20.8	7.9	19.8	7.4	24	20.2	30.5	24.5
SFA (g)	22.4	9.2	24.2	9.9	23.2	9.6	22.5	21.3	29.4	26.6
n-3 (mg)	57.4	23.4	59	22.7	58.1	23.1	NA	NA	NA	NA
n-6 (mg)	1707	882.7	1983.7	1011.1	1827	950.6	NA	NA	NA	NA

PUFA, Polyunsaturated Fatty Acid; MUFA, Monounsaturated Fatty Acid, SFA, Saturated Fatty Acid; HCS, Hunter Community Study; NA, data not available.

**Table 3 nutrients-11-00711-t003:** Unadjusted and adjusted linear regression between dietary intake of fatty acids and MMSE score in the HCS cohort.

Predictor Variable	Unadjusted RC	*p*-Value	Adjusted RC *	*p*-Value	Adjusted *R*^2^
Total fat (g)	−0.0027248	0.067	−0.0001	0.94	0.0495
Total PUFA	−0.0104	0.234	0.0014	0.898	0.0495
Total MUFA	−0.0083	0.052	−0.0028	0.611	0.0496
Total SFA	−0.0054	0.106	0.002	0.817	0.0495
n-3	−0.002	0.152	−0.0012	0.454	0.0497
n-6	−0.0001	0.051	0	0.787	0.0495

RC, regression coefficient; PUFA, polyunsaturated fatty acid; MUFA, monounsaturated fatty acid; SFA, saturated fatty acid. * Adjusted for BMI, smoking, age, education, dietary cholesterol, CHO and total sugar.

**Table 4 nutrients-11-00711-t004:** Unadjusted and adjusted linear regression results between dietary intake of fatty acids and total ARCS score in the HCS cohort.

Predictor Variable	Unadjusted RC	*p*-Value	Adjusted RC *	*p*-Value	Adjusted *R*^2^
Total fat	−0.0694	0.067	0.0301	0.673	0.004
Total PUFA	−0.0311	0.919	0.44	0.254	0.0046
Total MUFA	−0.0362	0.158	0.0434	0.822	0.004
Total SFA	−0.1712	0.144	0.0207	0.89	0.004
n-3	−0.0311	0.53	0.0359	0.528	0.0042
n-6	0.0018	0.126	0.0043	0.0004 **	0.0084

RC, regression coefficient; PUFA, polyunsaturated fatty acid; MUFA, monounsaturated fatty acid; SFA, saturated fatty acid. * Adjusted for BMI, smoking, dietary cholesterol, CHO and total sugar. ** Statistically significant (*p* < 0.008, adjusted to allow for multiple comparisons/hypotheses).

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
