# Peer review of "Dietary Long-Chain Fatty Acids and Cognitive Performance in Older Australian Adults"

_nutrients, 2019, doi:10.3390/nu11040711_

Round 1

Reviewer 1 Report

Introduction -

A very well written paper on fatty acids and cognition in older Australian. In the introduction there is an initial discussion of MedDiet and fatty acids, which is not rationale when considering the rest of the paper. The inclusion of the MedDiet would have been fine if there was analysis of the adhesion to the Mediterranean diet in this cohort. Authors need to rewrite the introduction with the discussion solely around the fatty acids and cognition. The Mediterranean Diet is actually considered low in n-6 fatty acids, another reason why your introduction is not rationale. The Mediterranean Diet is high in n-9 fatty acids, as well as having many other characteristic features, high in veg, green veg, fruit, low in meat and high in fibre. As such, this is somewhat of a long bow to draw here.  

Methods

line 123 - Please explain if this FFQ is sensitive enough for fatty acids, does it ask about different oils, spreads, salad dressings, nuts, seeds that are used.

Line 174 - fish oil intake was described and reported but what about other oil supplements (evening primrose oil etc?

Is there evidence to suggest the last 12 months is a reasonable period to assess dietary intake in relation to cognitive states?

Results

 Table 2 - missing digit for n-6 dietary intake for women (?) and men(?). Please check and confirm.

Discussion

lines 311-313 - here you state that n-3 FA intakes are not available from AHS results, but table 2 also indicates that n-6 results are not available - please rectify this and correct discussion as needed.

 Also is it reasonable to say that the dietary data from this study is representative of AHS if data on n-6 and n-3 is not available? Please consider and adjust.

Author Response

The authors of  “Dietary long-chain fatty acids and cognitive performance in older Australian adults” wish to kindly thank the reviewers for their time and input. Their insight has proven to be very valuable and the manuscript has been improved by the peer review process. We kindly ask that you view these changes included in the table below. 

Author Response to Reviewer Comments

Reviewer 1:

Reviewer comments:

Author Response

Introduction: A very well written paper on fatty acids and cognition in older Australian. 

Thank you for this comment. We are glad to receive this feedback.

In the introduction there is an initial discussion of MedDiet and fatty acids, which is not rationale when considering the rest of the paper. The inclusion of the MedDiet would have been fine if there was analysis of the adhesion to the Mediterranean diet in this cohort. Authors need to rewrite the introduction with the discussion solely around the fatty acids and cognition. The Mediterranean Diet is actually considered low in n-6 fatty acids, another reason why your introduction is not rationale. The Mediterranean Diet is high in n-9 fatty acids, as well as having many other characteristic features, high in veg, green veg, fruit, low in meat and high in fibre. As such, this is somewhat of a long bow to draw here.  

Thank you for your comment regarding the inclusion of the Mediterranean Diet (MeDi) within the introduction. The authors realise the rationale was not made clear and the manuscript has been edited to emphasize that while dietary pattern (including the MeDi which incorporates a variety of fatty acids), fish intake and fatty acid intake have all been shown to influence cognitive performance, the evidence is not conclusive as to which may be associated with cognitive performance. This provides the rationale that analysis of individual fatty acid intake may be beneficial to confirm. 

The abstract has been edited accordingly and the MeDi reference removed at; Page 1: Line 12.

“Convincing evidence exists for the positive effect of an improvement in diet quality on age-related cognitive decline”

The introduction has also been edited to clarify the rationale and to include that the Mediterranean diet has a lower proportion of n-6 fatty acids; Page 2: Line 51-57

“Epidemiological studies support the hypothesis that dietary quality and pattern can influence an individual’s cognitive performance throughout their life-cycle [7]. Fatty acid and fish intake in particular have been shown to have an effect on cognitive performance [12-15]. The Mediterranean Diet (MeDi) includes a wide range of foods and food groups, and therefore a range of fatty acids, including n-9 monounsaturated fatty acid (MUFA), n-3 and to a lesser extent n-6 polyunsaturated fatty acids (PUFA), and saturated fatty acids (SFA). It is the most studied dietary pattern to date and has been convincingly linked with lower rates of cognitive decline [8].”

The authors have also emphasized in the discussion that the n-6 FA significant result was unexpected in light of current evidence; Page 7: Line 274-278.

“Our findings demonstrated that individuals with higher consumption of n-6 FAs had better cognitive performance as measured by the ARCS. This is in contrast to what might be expected as many studies suggest that linoleic acid has an atherogenic role by increasing the oxidative modification of low-density lipoprotein cholesterol, a situation that would theoretically be detrimental to cognitive function [35,36].”

Methods:

Line 123 - Please explain if this FFQ is sensitive enough for fatty acids, does it ask about different oils, spreads, salad dressings, nuts, seeds that are used.

The FFQ used by the authors has been validated against four day weighed food records for most macro and micro nutrients (including total fat, saturated fat, monounsaturated fat and polyunsaturated fat) but not for GI or GL. 

Fatty acid intake was determined using an additional nutrient database as the NUTTAB database was determined as limiting due to data having too few decimal places. Use of fatty acid data from the Royal Melbourne Institute of Technology (Mann et al) ensured sensitivity for longer chain fatty acids but may have limitations in terms of assessing the biological availability of food sources. The authors acknowledge this may be a limitation of data collection and have included this in the discussion of the manuscript; Page 9: Line 356-358.

“The use of nutrient databases ensured sensitivity for FA intake but may be limited in assessing the bioavailability of food sources.”

Line 174 - fish oil intake was described and reported but what about other oil supplements (evening primrose oil etc?

Thank you for highlighting this. Detailed supplement intake was in fact, collected by the FFQ. This included whether participants used a dietary supplement and if so which brand, frequency and strength and therefore reflects the total additional supplement use by the participant and not just fish oil. The manuscript has been edited to reflect this change; Page: 4; Line 191.

“Self-reported total supplement use (including fatty acids such as fish oil) was also collected.”

Is there evidence to suggest the last 12 months is a reasonable period to assess dietary intake in relation to cognitive states?

Twelve months is a typical time frame for collection of dietary data using the FFQ methodology. Timeframes longer than this would be severely impacted by memory, especially in an older population. Additionally, timeframes shorter than this would be likely be inadequate to make any comparison with cognitive stateThe researchers believe that 12 months is appropriate to assess the habitual intake an older population in relation to cognition.

Results:

Table 2 - missing digit for n-6 dietary intake for women (?) and men(?). Please check and confirm.

Thank you for alerting us to this omission. The table has been rectified to include the missing digit. It now reads n-6 (mg) 1707.0; Page 6; Line 239-240. Table 2.

Discussion:

Lines 311-313 - here you state that n-3 FA intakes are not available from AHS results, but table 2 also indicates that n-6 results are not available - please rectify this and correct discussion as needed.

Thank you for pointing this out. The authors were trying to make a point in the discussion that was specific to n-3 fatty acids but it does make sense to report the omission of data in its entirety before making specific reference to the individual fatty acid. Please find the discussion changed at; Page 8: Line 331-333.

“When compared with the Australian Health Survey results, all obtainable FA levels were comparable, however LCn-3PUFA and LCn-6PUFA population intakes were not available for comparison [32]. As total PUFA intakes were similar, it has been assumed that n-3 FA and n-6 FA intakes would also be comparable.”

Also is it reasonable to say that the dietary data from this study is representative of AHS if data on n-6 and n-3 is not available? Please consider and adjust.

The authors believe that as the broader HCS intake data from across all macronutrients and specifically to total fat, PUFA, MUFA and SFA have been found to be comparable to the AHS data, then it may be assumed that this would also be the case for n-3 and n-6 fatty acid data. This assumption is highlighted in the manuscript; Page 8: Line 330-333

“When compared with the Australian Health Survey results, all obtainable FA levels were comparable, however LCn-3PUFA and LCn-6PUFA population intakes were not available for comparison [32]. As total PUFA intakes were similar, it has been assumed that n-3 FA and n-6 FA intakes would also be comparable.”

Reviewer 2 Report

Comments:

1 - I suggest to stratify the data between smokers and non-smokers. Smokers should be a bias in the final result and it is important to assess whether there is a relationship among non-smoker, smoker, diet, fatty acids, and cognition.

2 -  If the current study aimed to assess the associations between dietary long-chain fatty acids and cognitive performance, I suggest to evaluate the short-chain fatty acids (SCFAs) in the feces of theses patients and then correlate with cognitive performance. 

3 -  Discussion. ... in a large cohort of older Australian adults, aged 55-86 years (43.6% men and  56.4% women). Is 55 year-old considered a older person?

4 - What is the molecular mechanism that individuals with higher consumption of n-6 FAs presented better cognitive performance as measured by the ARCS? How do the fatty acids cross the blood-brain barrier? What are the receptors or mechanisms activated? Are there some mechanism that can activate or inhibit microglia or astrocytes cells in the brain?

5 - Is it possible to evaluate the effect of n-6 FA intake on separate cognitive domains?

6 - May you please  justify, why was not observed a positive associations between total n-6 FA intake and cognitive performance measured with the MMSE? What was the stratified data from Mini Mental State Examination (MMSE) between the participants? Are there association between SCFAs and mental state? 

Author Response

The authors of  “Dietary long-chain fatty acids and cognitive performance in older Australian adults” wish to kindly thank the reviewers for their time and input. Their insight has proven to be very valuable and the manuscript has been improved by the peer review process. We kindly ask that you view these changes included in the table below. 

Author Response to Reviewer Comments

Reviewer 2:

Comments:

1 - I suggest to stratify the data between smokers and non-smokers. Smokers should be a bias in the final result, and it is important to assess whether there is a relationship among non-smoker, smoker, diet, fatty acids, and cognition.

Thank you for this comment. The authors would like to clarify that smoking status was considered a potentially confounding variable and as such was accounted for within the regression modelling. The authors did not feel that the sample size included an adequate number of smokers to consider expanding this analysis further.

2 - If the current study aimed to assess the associations between dietary long-chain fatty acids and cognitive performance, I suggest evaluating the short-chain fatty acids (SCFAs) in the feces of these patients and then correlate with cognitive performance. 

While the authors agree such analysis would be an interesting addition to the manuscript, the current study is a secondary data analysis of a large cohort. Faecal samples were not collected from this cohort and therefore not available for analysis.

3 - Discussion. ... in a large cohort of older Australian adults, aged 55-86 years (43.6% men and 56.4% women). Is 55-year-old considered a older person?

Thank you for this comment. The authors believe while the consideration of 55 years old constitutes an ‘older’ segment of the population, this is subjective. However, a decline in cognitive performance can become evident from this age (and even earlier in some instances) and is often measured from this point. The cohort from which this study data were extracted had a mean age of 66.6 years with a SD of 7.6 years, meaning the majority of participants fell between the ages of 59 and 74 years (Page 5: Line 208-209: Table 1). The authors believe this justifies a starting age of 55 years.

4 - What is the molecular mechanism that individuals with higher consumption of n-6 FAs presented better cognitive performance as measured by the ARCS? How do the fatty acids cross the blood-brain barrier? What are the receptors or mechanisms activated? Are there some mechanism that can activate or inhibit microglia or astrocytes cells in the brain?

The authors of this cross-sectional analysis of secondary data were looking at the possible association between the dietary intake of different fatty acids and cognitive performance. Detecting an association between n-6 intake and better cognitive performance was a surprising result and this observational study would not presume to hypothesize a molecular mechanism as to how this might occur, merely that there was found to be an association which may warrant further investigation.

5 - Is it possible to evaluate the effect of n-6 FA intake on separate cognitive domains?

The effect of n-6 intake on separate cognitive domains was evaluated by the authors. It was presented in the results section; Page 7: line 259-264.

“When n-6 FA intake was tested against each of the five ARCS domains (Memory, Fluency, Visual, Language and Attention) there were no statistically significant associations for the Memory domain (RC= -0.000064; P= 0.889; R2= 0.0083), however, statistically significant associations were observed for the Fluency (RC= 0.0011432; P= 0.007; R2= 0.0057), Visual (RC= 0.0009889; P= 0.034; R2= 0.0050), Language (RC= 0.0010651; P= 0.047; R2= 0.0068) and Attention (RC= 0.0011605; P= 0.017; R2= 0.0099) domains.”

6.a. - May you please justify, why was not observed a positive association between total n-6 FA intake and cognitive performance measured with the MMSE? 

Thank you, the authors did note that there was a difference in the statistical significance of the two measures of cognitive performance. The authors provided a possible explanation in the discussion; Page 8: Line; 317-325.

“The MMSE is a well-recognised and commonly used cognitive measure in research, particularly survey-based research, however it is best used in a longitudinal setting as it is a good measure of cognitive change over time [42,43]. Due to the cross-sectional design of this study, the MMSE may not be sensitive enough to detect differences in cognitive performance between participants at one time point. It has also been suggested that the MMSE may only be sensitive when a patient is already significantly impaired, and therefore may not be suitable for the high-scoring healthy participants of the HCS [42]. However, the MMSE provided a comparison to the validated ARCS tool, which has been recently developed and shows greater sensitivity [31,44].”

6.b. - What was the stratified data from Mini Mental State Examination (MMSE) between the participants? Are there association between SCFAs and mental state? 

The authors agree that the addition of stratified data from the MMSE and SCFA intake would be an interesting addition to this study however, this was a secondary data analysis and as such the authors were not provided with that data to include in analysis here.

Round 2

Reviewer 1 Report

Thank you for the changes made in respect to the feedback that was given. I am satisfied with most of the comments.

However I am not confident that the authors can make the following statement: "When compared with the Australian Health Survey results, all obtainable FA levels were comparable, however LCn-3PUFA and LCn-6PUFA population intakes were not available for comparison [32]. As total PUFA intakes were similar, it has been assumed that n-3 FA and n-6 FA intakes would also be comparable."

n-6 and n-3 levels can vary considerably based on the level of fish consumption in the diet, the authors need to provide further proof or evidence for this statement.

Author Response

Author response;

Reviewer 1

Comment

Response

Thank you for the changes made in respect to the  Author response to reviewer;

Reviewer 1;

feedback that was given. I am satisfied with most of the comments.

However, I am not confident that the authors can make the following statement: "When compared with the Australian Health Survey results, all obtainable FA levels were comparable, however LCn-3PUFA and LCn-6PUFA population intakes were not available for comparison [32]. As total PUFA intakes were similar, it has been assumed that n-3 FA and n-6 FA intakes would also be comparable."

n-6 and n-3 levels can vary considerably based on the level of fish consumption in the diet, the authors need to provide further proof or evidence for this statement.

Thank you for your further comments and we appreciate your feedback.

The authors acknowledge that n-6 and n-3 levels can vary considerable based on the level of fish consumption in the diet. The limitations of available data mean further analysis is not possible. We will omit the phrase relating to inference of n-3 and n-6 intake.
